PERSPECTIVE

# Variation in Microbial Exposure at the Human-Animal Interface and the Implications for Microbiome-Mediated Health Outcome

Sahana Kuthyar,[a] Aspen T. Reese[a]

[a]Division of Biological Sciences, University of California San Diego, La Jolla, California, USA

**ABSTRACT**  The human gut microbiome varies between populations, largely reflecting ecological differences. One ecological variable that is rarely considered but may contribute substantially to microbiome variation is the multifaceted nature of human-animal interfaces. We present the hypothesis that different interactions with animals contribute to shaping the human microbiome globally. We utilize a One Health framework to explore how changes in microbial exposure from human-animal interfaces shape the microbiome and, in turn, contribute to differential human health across populations, focusing on commensal and pathogen exposure, changes in colonization resistance and immune system training, and the potential for other functional shifts. Although human-animal interfaces are known to underlie human health and particularly infectious disease disparities, since their impact on the human microbiome remains woefully understudied, we propose foci for future research. We believe it will be crucial to understand this critical aspect of biology and its impacts on human health around the globe.

**KEYWORDS**  One Health, animal, gut microbiome, human microbiome

Human-animal interfaces are a critical, but often overlooked, element of our environment and are comprised of our interactions with animals and, in turn, their microbes (1). These interactions encompass not only direct contact with animals, such as wildlife, livestock, and pets, but also sharing a common environment and contacting or consuming animal products. Across the globe, these interactions vary between populations and largely reflect industrialization and lifestyle transitions (2). Increasing anthropogenic disturbance and encroachment into wild habitats often intensify these interfaces or create new ones and, thus, increase the potential for microbial transmission (3, 4). Historically, much of the interest in microbial exposure at human-animal interfaces has focused on pathogenic microbes, especially zoonoses that can jump from animals to humans and lead to infection (3, 5, 6). Less often considered is the fact that microbial exposure also includes contact with commensal and/or beneficial microbes. As our appreciation for the role of the microbiome in shaping human biology grows (7–9), we may find that the spread of nonpathogenic microbes is equally important for understanding human health across human-animal interfaces.

The human gut microbiome reflects an individual's diet, genetics, lifestyle, and contact with the environment, all of which vary between and within human populations. Broad differences in microbiomes across populations have previously been attributed to differences in ecological factors, such as diet (10–14), antibiotic usage (15–17), and other lifestyle factors that change with industrialization (18). Variation in the human microbiome plausibly also stems from differences in environmental microbial exposure, particularly from interactions at human-animal interfaces (Fig. 1), but to date, these effects have received much less attention than other drivers. The diversity of human-animal interactions clearly contributes to distinct opportunities for microbial dispersal across interfaces (Box 1) and, thus, may shape microbiome states that are associated with particular human populations across the globe. This variation is important to understand, as it may underlie differences in microbial functioning that, in turn, contribute to population-level differences in human health (19, 20). As such, we

Address correspondence to Aspen T. Reese, areese@ucsd.edu.

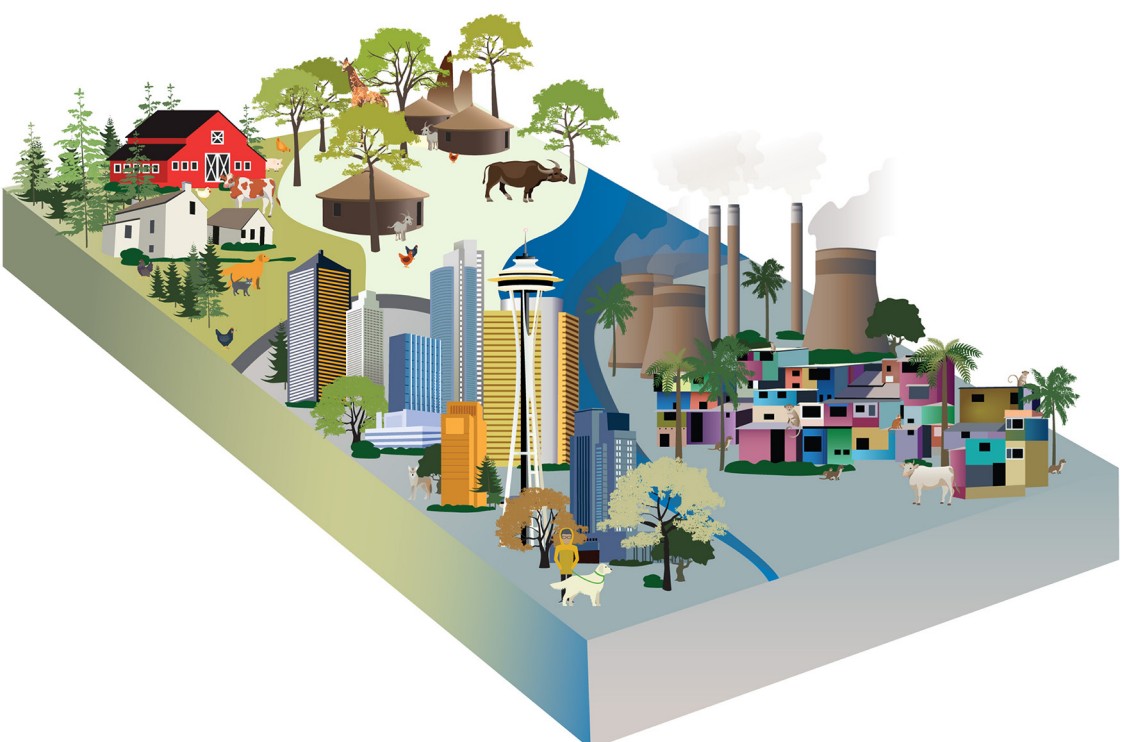

**FIG 1** Varied lifestyles promote differential interactions with animals, where not only the type of animal differs but so does the type of interaction. We hypothesize these distinct interactions shape microbial transmission across an environment and result in human gut microbiome variation. We illustrate some of the possible lifestyle contexts, including Western industrial rural (top left quadrant); non-Western, nonindustrialized, or traditional rural (top right quadrant); non-Western industrializing city (bottom right quadrant); and Western industrialized city (bottom left quadrant). Each setting will have a unique composition of direct and indirect interactions among humans, animals, and their shared environment. The animals shown here typify those possible interactions but are not a comprehensive representation. For example, in non-Western industrializing cities, such as Mumbai, some wild animals, especially monkeys, have adapted to living there. While wild animals, like coyotes, can be found in Western industrialized cities, they are different species, much less common, and less likely to interact with humans directly, resulting in a unique human-animal interaction suite between the two environment types.

need new and more extensive research to isolate the microbial effects of exposure to animals on human health relative to other lifestyle differences.

The diversity of the human experience belies the standard demarcation of human habitats as either industrialized or traditional, and it may be that intermediate lifestyles, such as those represented in populations undergoing market integration or living in urban centers in low- and middle-income countries, have the most to offer for understanding the complex factors that shape the microbiome and, in turn, human health. These intermediate populations can have a suite of nuanced behaviors and practices, where they are simultaneously exposed to industrialized elements and the natural environment, including animals and their microbes. For example, some humans within these intermediate populations may directly interact with wild animals (i.e., hunters [21]), while others may live near environmental reservoirs, such as rivers, where animal-associated microbes are encountered through drinking and washing. These routes expose populations to both pathogenic and commensal microbes, which may further shape their microbiome. Disentangling the microbiome effects of dichotomous microbial exposure at human-animal interfaces from other environmental and lifestyle factors could help explain and, thus, address global health disparities. While the details of these disparities are beyond the scope of this paper, other recent reviews provide extensive detail on them (e.g., 20, 21).

Here, we utilize a One Health framework, jointly considering humans, animals, and their shared environment, to describe microbial exposure along global human-animal interfaces and its potential implications for human health. We first review the environmental factors that drive microbial variation across human populations and discuss human-animal interfaces as another ecological mechanism that shapes differences in microbial exposure. We

then specifically outline the functional consequences that may come from interspecific microbial transmission. While microbial exposure at the human-animal interface includes both pathogens and commensals, much of the existing literature and, thus, many of the examples in this essay, is focused on pathogens. In some cases, we can extrapolate findings from pathogen research to commensals and from animal studies (e.g., domesticated animals) to humans. Altogether, the impact of human-animal interfaces on the human microbiome is currently understudied, but we believe it will be crucial if we aim to understand this critical aspect of biology and its impacts on human health around the globe.

---

**BOX 1: PATHWAYS OF MICROBIAL TRANSMISSION ACROSS HUMAN-ANIMAL INTERFACES**

Increased interactions between humans, animals, and the environment can lead to a greater volume and diversity of microbes moving between different host species. Although much of what we know about transmission routes relies heavily on pathogen data, it is probable that commensal microbes are transmitted through these routes as well, although perhaps at lower rates (22, 23). Microbes, including pathogens like rabies virus and *Mycobacterium tuberculosis*, can be directly transmitted via contact with wild animals (24, 25). Domesticated animals, including livestock and pets, can harbor their own pathogens (26, 27) and also serve as bridges that facilitate microbial transmission at human-animal interfaces. For example, the Nipah virus emerged when farm pigs ate fruit contaminated by bats and spread the virus to nearby humans (28).

The environment (e.g., soil, water, and air) may also act as a vector that facilitates microbial transmission across human-animal interfaces. Fecal and manure runoff from farms have previously been linked to the emergence of zoonotic infectious disease in humans (26, 29–31) and wildlife (32–34). Further, the causative agent of Q fever, *Coxiella burnetii*, spreads via wind from livestock reservoir hosts to nearby human populations (35). Spore-forming *Coccidioides immitis* can also persist in the soil and infect humans who are exposed to such soils (36). In addition to these abiotic routes, the human environment and cultural norms can also promote transmission. Animal markets contain high densities of humans interacting with animals, both live and carcasses, which can lead to novel pathogen exposure, as seen when contact with animals in live poultry markets led to the transmission of H5N1 (37). Consumption of undercooked meat and other animal products that harbor live microbes can also serve as routes for microbial host jumping (e.g., hepatitis E virus [38]). Hand-washing practices after contact with animals and animal products can mitigate the effects of microbial exposure, but these, along with other hygienic practices, vary between human populations (39).

Novel direct and indirect transmission of microbes between animals and humans is expected to increase as hosts move across the globe with growing trade and travel. Microbes may be brought in when pets and exotic wild animals are imported for personal use or commercial trade (40, 41). Monkeypox was introduced to North America with the import of African Gambian giant rats (42) and, on multiple occasions, H5N1 virus has been detected in birds imported to Europe (41, 43).

It is worth noting that the health effects of microbes can change when they jump to a novel host or appear in a new region (44). As such, predicting the outcome of transmission, including rates of microbial spread, length of colonization, and virulence (45), may be difficult. Microbes can become more pathogenic in new hosts. For example, direct human-animal contact led to the zoonotic transmission of SIV from chimpanzees and sooty mangabeys to humans (5), with human immunodeficiency virus as a much more virulent virus in humans than simian immunodeficiency virus is in nonhuman primates. However, pathogens can also become less virulent when they jump to a new host, such as when hematopoietic necrosis virus spread from sockeye salmon to rainbow trout (46, 47).

mSystems®

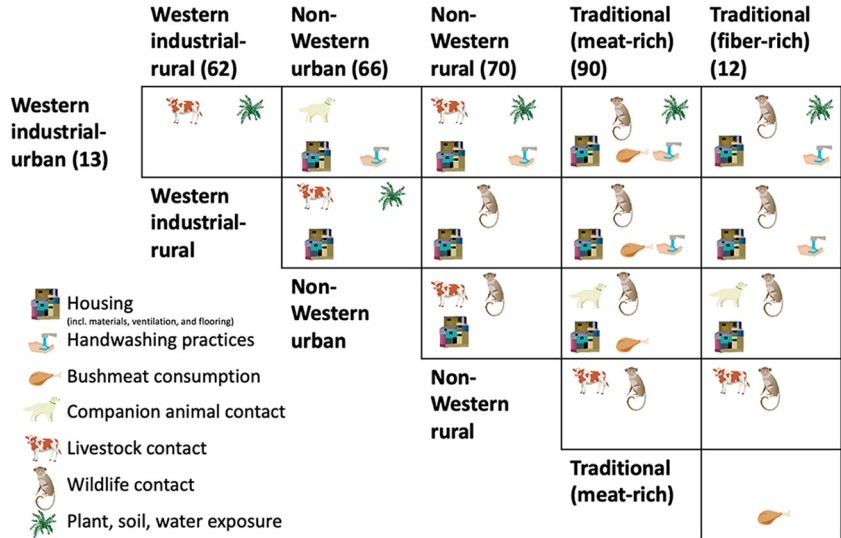

**FIG 2** Human-animal interface variables anticipated to impact microbial communities across human populations. Citations are for a gut microbiome study analyzing a representative group for that population type but do not necessarily evaluate how human-animal interfaces shape those microbiomes. The matrix highlights critical variables that may cause differences between two populations (e.g., consumption of bushmeat is an important difference between Western industrial-urban and traditional meat-rich populations).

## VARIATION IN HUMAN MICROBIAL COMMUNITIES ACROSS POPULATIONS

The human gut microbiome is shaped by an amalgamation of ecological variables, including diet (10, 13, 48), delivery mode (49), antibiotics (16, 17), and environmental microbial exposures (50–52), which are not consistent across human populations (50, 53). Such differences are stereotypically illustrated through broad comparisons between rural non-Western and industrialized Western countries (12, 14, 54, 55). Those comparisons show that the former tend to have distinct composition and higher microbial diversity than the latter (e.g., Burkina Faso versus Italy [14]; the Hadza versus Italy [12]; Papua New Guinea versus the United States [55]). Some rural and urban populations within non-Western countries have also been found to display microbiota differences that result from variation in lifestyle factors (56–58). For example, in urban Manaus, Brazil, increased sanitation, hygiene, and time spent within built environments precluded gut microbial colonization by environmental microbes from soil, arthropods, and plants (59).

One variable that is rarely considered explicitly but may contribute to inter- or intra-population microbiome variation is microbial exposure specifically arising from human-animal interfaces (Fig. 2). Such an effect is intuitive, since animal exposure is a known route of microbial transmission and human-animal interactions with animals vary across populations (Fig. 1). As the human microbiome reflects the built environment (50), it is also likely to reflect the natural one, and some recent studies support this hypothesis. Within Dutch families, which otherwise are expected to largely share their microbiota, significant differences in gut community and functional gene composition have been found between farm and meat processing plant workers and their household members who do not work with livestock (60). Similarly, Chinese swine farmers show distinct microbiome composition from nonfarmer villagers (61). Such microbial impacts of animal contact can manifest rapidly: veterinarians who started working with farm animals quickly picked up pathogenic taxa and antibiotic resistance genes from the animals under their care and then lost at least some of them after ceasing work on the farm (62).

We expect variation in human-animal interface effects to be particularly noticeable in intermediate human populations. For example, people in developing economies may work in industrial settings (i.e., factories) but still reside in environments more intertwined with the natural world (i.e., using untreated water sources or having less effective

ventilation systems in their homes). Alternatively, rural populations in Westernized areas may regularly interact with livestock yet eat Western diets and live in homes made of industrial materials and maintained with high hygiene levels. These idiosyncratic lifestyle combinations may underlie population-specific microbial compositions that are distinct from the stereotypical highly industrialized or traditional populations.

Unfortunately, historically relatively few samples have been collected in intermediate populations, such as those in urban slums in developing megacities or agricultural communities beyond pure subsistence. The limited data that are available for these intermediate populations, such as individuals living in peri-urban towns in Peru and South Africa as well as an Irish indigenous group transitioning away from nomadic living, exhibit gut microbial signatures from both highly industrialized and traditional populations but also display features of their own (16, 63, 64). Similarly, a recent worldwide survey found gut microbial composition clustered by lifestyle, with urban nonindustrialized and rural industrialized falling between the more stereotypical urban industrialized and rural non-industrialized populations (65). These intermediate microbial signatures are typically ascribed to lifestyle factors, and while human-animal interactions are not always explicitly measured, they may be reflected in the factors found to be significant. For example, across an urbanization gradient from a remote Amerindian village to urban Manaus, significant microbial variation was tied to urbanized housing, which is associated with reduced environmental, and potentially animal, exposure and increased exposure to industrial building material and antimicrobial cleaning products (66). Similarly, housing conditions best predicted the composition and diversity of gut microbial communities in both Irish indigenous communities and Ecuadorian populations undergoing market integration (63, 67). Other studies in Himalayan and Hadza populations (54, 68) found that drinking water and cooking method, both routes of direct microbial exposure and proxies for environmental and animal exposure, were associated with differences in gut microbial composition. However, it is worth noting that the highlighted variables in all these studies typically only explain approximately 30% of variation in microbial composition. It may be that impacts of microbial exposure at the human-animal interface are embedded partially in that 30%, or, if they were explicitly tested for, such impacts could play a role in explaining some of the remaining 70% of variation. Unfortunately, most studies of intermediate populations use methods that describe composition but not function, and none are able to ascribe a specific proportion of microbiome variation across populations strictly to human-animal interactions.

Insofar as we care about microbiome composition across populations, it is because differences can have implications for microbial functioning and, in turn, host health. The gut microbiome has recently been proposed as a pathway that connects an individual's environment to health disparities (69). As such, microbial exposure from human-animal interactions may help explain health disparities around the globe, both through microbiome effects as well as microbial transmission from animals (i.e., zoonotic disease risk) (19, 20, 70). To date, though, public health studies that do consider environmental and animal exposure only highlight the risks posed by zoonotic pathogen transmission at human-animal interfaces and fail to describe broader microbial patterns (71). As such, it is hard to disentangle if health disparities are occurring due to exposure to pathogenic microbes and resulting zoonotic disease from human-animal interactions or decreased exposure to appropriate commensal microbes due to reduced and/or altered human-animal contact. Nevertheless, other lines of evidence suggest important health effects of interactions between humans and animals.

## CHANGES IN HUMAN HEALTH DUE TO VARIATION IN MICROBIAL EXPOSURE

We expect changes in microbial exposure from human-animal interfaces arising from direct animal contact or via a shared environment to shape the microbiome and, in turn, to contribute to differential human health across populations. This thought echoes, but broadens, the hygiene hypothesis, which posits that since microbial exposure early in life drives immune development and training (72) and promotes

relationships with coevolved commensals that aid in human immunomodulatory functions (73), reduced microbial exposure due to heightened hygiene would have a negative impact on health, a pattern particularly noticeable in high rates of allergic and autoimmune disease. We argue the effects of microbial exposure are broader than just immune system training. Moreover, there are trade-offs associated with microbial exposure during human-animal interactions, as one can be exposed not only to commensal or beneficial microbes but also to potential pathogens. So, while exposure to microbes early in life can prime the immune system or provide metabolic functions, without other measures to combat infection (i.e., hygiene and sanitation), such exposure can also lead to childhood morbidities and mortalities. There are many such health outcomes and potential disparities that result from differences in microbial exposure (69); we focus here on the following: specific commensal and pathogenic microbial exposure, changes in colonization resistance and immune system training, and the potential for other microbial functional shifts.

**Dichotomous microbial exposure.** As transmission routes for pathogenic and beneficial microbes likely overlap, people are exposed to a spectrum of microbes at human-animal interfaces; thus, we must consider both risk and protective factors with respect to the resulting impacts on health. Although human populations in industrial settings have greatly reduced infectious disease burdens relative to very low-income populations through increased hygiene and sanitation, they still have high potential for pathogen exposure from animals. Interacting with livestock is associated with microbiome differences (60, 74), and working on industrial farms is a known risk for zoonotic transmission and transmission of pathogens with antimicrobial resistance (75, 76). Indeed, the highest risk of pathogen spillover occurs in areas where large, intensified industrial livestock units exist in close proximity to family-owned, small-scale farms and degraded, natural habitats (77). Even in rural agricultural communities, animal feces has been identified as a potent potential avenue for microbial gene transfer to human-associated microbes (16). Equally important, though, is how interactions with livestock may provide routes for beneficial microbial exposure that positively impact human health. For example, the household microbiome of small-scale pig farmers harbored increased microbial diversity compared to that of suburban homes (78).

Interactions with domesticated companion animals or their feral relatives are likely additional routes of microbial exposure that differ across populations. Dogs are known reservoirs for *Toxocara* (79), and puppies, in particular, can serve as transmission vectors for *Campylobacter jejuni* to humans (80). Similarly, cats are the natural host for *Toxoplasma gondii*, and transmission of this pathogen between cats and pregnant women can lead to congenital toxoplasmosis, which can severely impact fetal development and physiology (81). Pets may also be a viable source of beneficial microbial exposure (82) and have even been proposed as a microbiome-based therapy (83). Certainly, studies have shown that people share commensal and beneficial gut and skin microbes with their dogs (82, 84), although another comparison between adults with and without pets only found differences in the abundance of seven microbial taxa and did not find any differences in microbial diversity or overall composition (85). In infants and young children, exposure to household pets is associated with increased gut microbial diversity (86, 87). However, it is unclear if interactions with dogs actually transmit dog-associated or environmental microbes to humans or if dogs merely serve as a conduit for microbial sharing between humans.

The ways in which humans consume animals and their products can also serve as a potential route for commensal and pathogenic microbial exposure. Handling and consuming raw or undercooked meat, including bushmeat from wild animals, across different geographical areas is commonly associated with zoonotic infection (88). Inuits often consume raw meat (89), and an experimental study that mimicked Inuit traditions of preparing seal meat found that the pathogen *Trichinella nativa* was still prevalent after partial cooking (90). Within populations, some individuals, such as workers

whose occupations revolve around meat (e.g., butchers and meat inspectors), are more likely to share microbes with meat products and may also be at a higher risk for pathogen exposure than others (60).

**Changes in colonization resistance and disrupted immune training.** Beyond potential exposure, the realized health outcomes of being exposed to commensal and/ or pathogenic microbes will most likely differ between populations due to variation in interactions at human-animal interfaces. In particular, we expect to see impacts on gut colonization resistance, where the microbiome and host immune system both play an important role in inhibiting colonization and growth by pathogens as well as promoting the growth of nonpathogens (91). Microbiome-mediated colonization resistance serves as a first line of defense (92–94), and local microbial exposures, such as those from animals, will determine how effective this defense is. Differential microbial community composition may drive and/or be driven by specific recognition and defense against invading microbes (95) or by indiscriminate competitive interactions between the existing community and invaders (96). Apart from barring pathogens from colonizing, a diverse microbial community can also mitigate clinical symptoms of infection (92, 93, 97). While there is no evidence currently available showing altered colonization resistance between human populations due to animal exposure, other ecological disturbances (such as antibiotics treatment [98]) are known to reduce colonization resistance.

Host immune activity can also be impacted in human populations who experience altered microbial exposure, especially as early life microbial exposure primes T-cell production and function (99) and trains the immune system to tolerate commensal microbes and resist pathogenic microbes (100). These mechanisms are implicated in poor adaptive immune system development observed in industrialized human populations with decreased environmental exposure (101, 102). Altered exposure patterns could also result in decreased immune-mediated colonization resistance, which would allow pathogens or cheating strains of beneficial microbes to exploit immune weaknesses (103, 104). These differences may not be obvious in observational studies, as industrialized populations are expected to experience reduced pathogen exposure due to heightened hygiene, but we predict that given the same challenges arising from proximity to animals and animal products, people in these industrialized populations would be especially susceptible. Worryingly, these impacts may be compounded across generations, leading to increased rates of immune dysfunction within populations (102, 105).

If a diverse microbial environment is needed for lifelong human health (106), it is tempting to speculate that humans with early-life contact with animals will have more effective immune systems due to increased exposure to animal-associated microbes, either from wild or domestic animals or environmental substrates. However, these people will still have high exposure to pathogenic microbes through the same routes, potentially outweighing the strong priming of their immune systems. Reliance on antimicrobials to combat pathogens can have off-target effects on commensals (107, 108), further complicating our understanding of how the environment shapes host health. Moving forward, plans to improve the quality of public health across the gradient of human-animal interfaces will need to intentionally approach supporting benefits and preventing costs of microbial exposure.

**Potential functional shifts.** The microbiome plays a crucial role in shaping facets of human physiology and development beyond immune function, including metabolism (8) and behavior (9). As microbial composition and diversity vary across populations, so might microbial function and, in turn, these other host phenotypes. For example, human populations have different abundances of genes encoding short-chain fatty acid (SCFA) production represented in their gut microbiomes (109). SCFAs are taken up for host usage, shaping metabolism and broader health status (110, 111). For instance, a study in multiple countries in Western Europe, which found a link between microbial exposure on farms and protection against asthma, proposed increased levels of butyrate, a microbially produced SCFA, as the mechanism (79). Another study in Sweden correlated high concentrations of valeric acid, another SCFA, with low rates of eczema in farm children compared to rural children not raised on farms (80). Since

microbial composition and functional potential vary globally, it is likely we can expect similar variation in other microbially mediated host functions like metabolism and development, even if studies have not yet evaluated any differences.

How microbially mediated functions differ across human-animal interfaces is currently unknown. Allergies and other autoimmune diseases are heightened in populations with low microbial exposure early in life (112, 113), and previous studies have found correlations between increased exposure to pets and decreased rates of atopic disease and allergies (114–116). Owning a pet may also have protective properties against microbial dysbiosis, irritable bowel syndrome, and metabolic disease (87, 117, 118). As such, it is hard to imagine that more widespread effects ascribable to interactions with animals are not present across populations.

## EVIDENCE FROM DOMESTICATED ANIMALS

Domesticated animals may be apt models to study variation in microbial exposures and resulting host health disparities along human-animal interfaces, providing lines of evidence above and beyond those from observational human studies. Domesticated animals have an ecological relationship with humans and are already considered models for human biomedical research (119). Furthermore, domesticated animals, such as livestock and pets, along with urban-adapted animals, represent a breadth of environmental and lifestyle settings that emulate what is seen in human populations across the globe. Domestication itself is associated with shifts in gut microbial communities across multiple mammalian host species (120–122), and the microbial changes resulting from animal domestication parallel those from human industrialization (121). Just as humans experience shifts in factors that impact the microbiome during industrialization, so do domesticated animals experience new diets, altered social structures, and different housing and environmental exposure relative to their wild progenitors. Importantly, there are parallels in how humans and domesticated animals interact with other animals across the gradient of industrialization. Wild animal contact is likely higher for both humans and domesticated animals in less industrialized contexts, whereas more industrialized and domesticated humans and animals likely interact with more companion animals. These differences will impact the frequency and diversity of microbial transmission via human-animal and animal-animal interactions. As such, domesticated animals are an especially promising model for understanding human health outcomes that result from variable microbial exposures across the gradient of industrialization.

Similar to what is seen in humans, the physical environment is an important factor that shapes domesticated microbial communities via microbial exposure. Farm animals living in free-range, organic environments have different microbial exposures and different microbiomes than those in commercial, factory environments (123, 124). For example, piglets born indoors have been shown to have different microbial compositions, lower microbial diversity, and increased expression of mucosal innate genes compared to piglets born outdoors (125). However, raising piglets in a high-hygiene facility, which limits microbial exposure, diminished the differences in microbially mediated immune function between indoor- and outdoor-born piglets (126). Environments where farm animals are housed in confined spaces but not kept under hygienic conditions are suggested to provide ideal conditions for the rapid spread of microbes (41, 127). To what extent these rapidly spreading microbes are commensal and/or pathogenic as well as their impact on host health remains to be explored.

The physical environment also shapes microbial exposure and the resulting host microbiome in companion and lab animals. For example, dogs living in suburban habitats harbor lower microbial diversity compared to rural and urban dogs (128). Rural and urban dogs may be exposed to different environmental substrates (i.e., soil-, plant-, or animal-associated microbial sources for rural dogs versus urban-associated microbes from the built environment and human skin in urban dogs), whereas suburban dogs may have very low exposure to environmental substrates and conspecifics (129). It is possible that, across the urbanization gradient, pets that experience altered microbial

exposures develop health problems, including allergies, chronic diarrhea, and inflammatory bowel disease (129, 130). Additionally, moving mice outdoors alters their gut microbiome composition and their immune function in part due to gut microbial changes (131, 132). Early life microbial exposure, whether from natural sources or experimental treatment, has been shown to shape the microbiome and in turn host immune function in lab mice, with long-lasting fitness implications (133–135).

In the wild, inter- and intraspecific microbial transmission occurs between individuals living near each other and can result in microbial composition convergence (136, 137). This phenomenon has also been demonstrated in lab models, such as mice and zebrafish, where cohousing facilitates the horizontal microbial transmission between individuals (138, 139). While there are opportunities for microbial dispersal in mixed herds of domesticated animals, relevant data currently do not exist for nonlaboratory contexts (i.e., companion and agricultural animals). It is likely, however, that distinct species of animals living on a given farm have more microbial community composition similarities with each other compared to animals living on different farms, especially as environmental samples taken across multiple farms can reflect distinct microbial signatures (140).

Factors that change with industrialization, including population density, social networks, diet, and exposure to green space, may also impact gut microbial communities of animals that have adapted to the human environment even without being subject to artificial selection (52, 141, 142). For example, urban land cover strongly predicts microbial community composition and was associated with decreased microbial diversity and higher *Salmonella* prevalence in American white ibises (143). Similarly, gulls living in more urban landscapes have less diverse gut microbiomes (144), and urban house sparrows harbor lower microbial diversity and fewer metabolic functions than rural sparrows (145). While these urban-adapted animals are not considered domesticated animals, they still interact with humans in a way that may shape their microbiomes and may, in turn, impact or inform human populations.

## FUTURE DIRECTIONS

These ideas augment previous work describing microbial-driven differences between human populations and propose new foci for future research. While there is much work still to be done, we hypothesize that different interactions with animals contribute to shaping the human microbiome across populations. Microbial transmission at human-animal interfaces is of critical importance for emerging zoonotic pathogens, so understanding variation in human microbial exposures across populations is important for modulating disease risk, especially in developing countries. As it is impossible to avoid pathogens altogether, using a One Health, microbial perspective for human health interventions will be salient, especially when beneficial microbial exposure may mitigate the health outcomes from pathogenic infection.

As variable microbial exposures and accompanying health disparities across populations go past crude level assessments of industrialization, data from highly Western and urbanized centers (10, 12) and traditional societies (48, 54) may not be generalizable to other human populations across the gradient of industrialization (63, 66, 68, 146). Compared to traditional populations, Western industrialized microbiomes broadly present a loss of microbial diversity and commensal-associated metabolic functions (105), which is conventionally associated with negative health outcomes (101, 147). However, we need to measure host phenotypes via anthropometrics and survey data across many more populations to assess if and when increased microbial diversity or specific microbial compositions correlate with increased microbial functioning and if these functions manifest in positive host health outcomes. Without analyzing microbial function in intermediate populations as well, we fail to consider the health disparities they face in being exposed to a wide range of microbes. We also need to acknowledge the biases regarding which populations are targeted in microbiome research and recognize that our current understanding of microbially mediated health may not be complete. As such, sampling diverse, underrepresented populations in

future studies is crucial to capture the range of microbiomes across the human species and to specifically recognize the large variation of human-animal interface effects on the human microbiome (146, 148). One route to doing so is leverage ongoing One Health studies to characterize the gut microbiome of people for whom fecal samples, health information, and environmental covariate data are currently available. Here, using already banked samples and implementing the practice of banking future samples will be crucial to gain access to data from a larger representation of human populations. Furthermore, utilizing field-friendly techniques such as storing fecal samples in ethanol may open avenues for more research on human-animal interface effects in locales where freezer availability is limited. By considering rapidly industrializing populations, we can assess how recent changes in human-animal interactions along the gradient of industrialization may drive microbial spread and shifts in populations.

In addition to analyzing existing human samples, we should also focus on collecting samples from nonhuman hosts (i.e., animals, water, and soil) to better understand how pathogenic and commensal microbes cycle between humans, animals, and the environment. Creating social or interaction networks to map microbial transmission at human-animal interfaces could identify humans, animals, or environmental features that act as microbial superspreaders. Collecting samples multiple times within a year will be important for identifying seasonal shifts in microbial exposure from host-associated and environmental microbial communities (149, 150).

Apart from sampling populations at a broader geographic and temporal scale, we could also benefit from using domesticated animals as observational comparisons and experimental models. Experimental manipulations with domesticated animals could help disentangle microbiome effects, and the resulting host impacts, of animal-associated microbial transmission from other environmental and lifestyle factors. These manipulations may also aid in determining the direction and dynamics of microbial transmission during host species interactions, potentially allowing the creation of predictive models of beneficial and pathogenic microbial transmission (51). Domesticated animals can be considered not only as examples for disparities in microbial exposure but also as leading players in future efforts to mitigate the health effects of altered microbial diversity in industrial human settings. Importantly, domesticated animals may serve as microbial reservoirs in otherwise depauperate environments and mitigate microbially mediated diseases. Having domesticated animals around us may not be the only viable microbial reservoir, however, as other interventions into the built environment and health care will be important routes for improving microbial function across populations.

These outstanding questions as well as other points brought up in this article can help direct transdisciplinary future research into understanding the nuances of how microbial exposure across human-animal interfaces influences health. Identifying what factors drive microbial variation and its resulting health impacts will provide basic scientific knowledge, direct the design of useful microbiome-targeted manipulations, and inform human decision-making from the scale of individual behavior through to national and international policy. Microbial transmission at human-animal interfaces has often been viewed as a threat, but by better understanding its effects on the microbiome, we may yet find it can be an opportunity for good as well.

## ACKNOWLEDGMENTS

We thank Jessica Diaz, the Jackrel lab, and the two anonymous reviewers for insightful feedback for this article. We also thank Suzanne Ishaq for the invitation to contribute to the Microbes and Social Equity special collection.

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
