## [Reviewer comments · mSystems]

Variation in microbial exposure at the human-animal interface and the implications for microbiome-mediated health outcome

Sahana Kuthyar and Aspen Reese

Corresponding Author(s): Aspen Reese, University of California, San Diego

Review Timeline:

Submission Date:	May 10, 2021
Editorial Decision:	June 17, 2021
Revision Received:	July 8, 2021
Accepted:	July 21, 2021

Editor: Suzanne Ishaq

Reviewer(s): Disclosure of reviewer identity is with reference to reviewer comments included in decision letter(s). The following individuals involved in review of your submission have agreed to reveal their identity: Aashish Jha (Reviewer #2)

Transaction Report:

DOI: <https://doi.org/10.1128/mSystems.00567-21>

June 17, 2021

Dr. Aspen T. Reese
University of California, San Diego
La Jolla, CA

Re: mSystems00567-21 (Variation in microbial exposure at the human-animal interface and the implications for microbiome-mediated health outcome)

Dear Dr. Aspen T. Reese:

Thank you for submitting your manuscript to mSystems. We have completed our review and I am pleased to inform you that, in principle, we expect to accept it for publication in mSystems. However, acceptance will not be final until you have adequately addressed the reviewer comments.

The authors and I agree that this is a well composed manuscript, and only minor suggestions have been made.

Thank you for the privilege of reviewing your work. Below you will find instructions from the mSystemseitorial office and comments generated during the review.

Preparing Revision Guidelines

For complete guidelines on revision requirements, please see the Instructions to Authors at <https://msystems.asm.org/sites/default/files/additional-assets/mSys-ITA.pdf>. **Submissions of a paper that does not conform to mSystems guidelines will delay acceptance of your manuscript.**

Sincerely,

Suzanne Ishaq

Editor, mSystems

Journals Department
Reviewer comments:

Reviewer #1 (Comments for the Author):

This paper utilizes a One Health framework to examine the range of variation in human-animal interactions that influence human health. While providing a primarily theoretical summary, the paper provides direction for promising avenues for hypothesis generation and future research. Importantly, the paper moves beyond over-simplified dichotomies of western urban industrialized populations vs rural developing populations, and provides a helpful figure (Fig 2) which hypothesizes variables that may differ between 5 different settings or types of population (defined by industrialization, urban-rural, and meat vs fiber basis). The authors argue that human-animal interactions are a key component of the environment, and influence health disparities. Thus, the paper serves to link an important aspect of environmental influences on the microbiome with a primary focus of public health, health disparities.

The authors lay out pros and cons of human-animal exposures, and discuss questions about the relative contribution to variation in health outcomes, raising many interesting questions for research along the way. The emphasis that intermediate lifestyles along the traditional-industrialized continuum may provide more to offer for understanding complex factors shaping the microbiome and human health is important, as much literature has focused on extremes of western urban industrialized populations compared to hunter-gatherers like the Hadza. One of the biggest contributions of the paper is the argument in favor of future research on intermediate populations, such as those in urban slums or agricultural communities. and especially considering housing effects.

Suggestions:

The paper raises questions, for example about farming practices and zoonotic infectious disease - thus raising issues of behavior and policies, which could be mentioned.

While hygiene is mentioned in the text several times, and two references specifically address the hygiene hypothesis, the text does not mention the hygiene hypothesis. This omission seems important and should be rectified. (see particularly p. 8-9)

The authors argue that typically-studied variables only explain 30% of variation in microbial composition, but what % of variation in health disparities do the authors think human-animal interactions explain? (P. 6)

There are likely tradeoffs associated with human-animal exposure, of positive immune system training with exposure to pathogens (p. 9). I wonder if a clearer description (figure?) of the tradeoffs might be helpful to readers? Perhaps incorporated somehow into Fig 1 - which is a pretty figure, but does not add that much beyond the text. Fig 2 is more useful in terms of generating hypotheses. Why do domestication and industrialization have similar shifts in microbial communities? Due to diet, housing, multispecies or other environmental interactions, stress? Since the authors are aiming to promote research in this important area, some hypotheses about factors influencing the similarities would be welcome. (mentioned on p10 and then factors are proposed on p.12 - perhaps connect them more closely in the text)

The discussion of suburban as opposed to rural-urban was interesting, and highlights environmental variation and exposure issues.

Reviewer #2 (Comments for the Author):

I had the pleasure to read the manuscript by Kuthyar and Reese titled "Variation in microbial exposure at the human-animal interface and the implications for microbiome-mediated health outcomes." In this manuscript, the authors implement a One Health framework to explore the how human-animal interactions influence human health via changes in the gut microbiome.

This paper is well written, thoughtful, and the authors describe various aspects of human-animal interface that has the potential to influence human microbiota by altering pathogenic as well as commensal microbes, which may influence human health.

I do not have major changes to recommend. I have some minor comments which I provided using track changes option in the attached word document.

We thank the reviewers for their comments. We have submitted a revised manuscript document with track changes highlighted and a clean copy. The line numbers cited below refer to the clean manuscript version. Reviewer comments are copied in *italics* with our response following. New text is shown in **green here.**

Response to Reviewers

Reviewer 1 Suggestions:

RIC1: The paper raises questions, for example about farming practices and zoonotic infectious disease - thus raising issues of behavior and policies, which could be mentioned.

The reviewer is right that human-animal interface effects could go beyond basic microbiological understanding to have implications for human behavior and policies. We now discuss the concept that differences in behavior within and between populations could impact microbial exposure and resulting health impacts with more detail in Lines 55-60 and that studying human-animal interface effects on the microbiome could inform policy in Lines 437-440.

Lines 55-60: For example, some humans within these intermediate populations may directly interact with wild animals (i.e., hunters (21)), while others may live near environmental reservoirs such as rivers where animal-associated microbes are encountered through drinking and washing. These routes expose populations to both pathogenic and commensal microbes, which may further shape their microbiome.

Lines 437-440: Identifying what factors drive microbial variation and its resulting health impacts will provide basic scientific knowledge, direct the design of useful microbiome-targeted manipulations, and inform human decision-making from the scale of individual behavior through to national and international policy.

RIC2: While hygiene is mentioned in the text several times, and two references specifically address the hygiene hypothesis, the text does not mention the hygiene hypothesis. This omission seems important and should be rectified. (see particularly p. 8-9)

While the Hygiene Hypothesis is one of the more recognized concepts in the microbiome field, the reviewer is correct that an explicit definition would be helpful to ensure all readers are on the same page. We now explicitly discuss the Hygiene Hypothesis and its relationship to our ideas on Lines 193-201 before delving into the specific changes in human health as a result of microbial exposure.

We expect changes in microbial exposure from human-animal interfaces arising from direct animal contact or via a shared environment, to shape the microbiome and, in turn, to contribute to differential human health across populations. This thought echoes, but broadens, the Hygiene Hypothesis which posits that since microbial exposure early in life drives immune development and training (73), and promotes relationships with co-evolved commensals that aid in human immunomodulatory functions (74), reduced microbial exposure due to heightened hygiene would

Variation in microbial exposure at the human-animal interface and the implications for microbiome-mediated health outcome

have a negative impact on health particularly noticeable in high rates of allergic and autoimmune disease. We argue the effects of microbial exposure are broader than just immune system training.

RIC3: The authors argue that typically-studied variables only explain 30% of variation in microbial composition, but what % of variation in health disparities do the authors think human-animal interactions explain? (P. 6)

A fair question, but one that is impossible to answer without more data. We have added text that considers this question more explicitly, though, in Lines 171-178.

However, it is worth noting that the highlighted variables in all these studies typically only explain approximately 30% of variation in microbial composition. It may be that impacts of microbial exposure at the human-animal interface are embedded partially in that 30%, or, if they were explicitly tested for, such impacts could play a role in explaining some of the remaining 70% of variation. Unfortunately, most studies of intermediate populations use methods that describe composition, but not function, and none are able to ascribe a specific proportion of microbiome variation across populations strictly to human-animal interactions.

RIC4: There are likely tradeoffs associated with human-animal exposure, of positive immune system training with exposure to pathogens (p. 9). I wonder if a clearer description (figure?) of the tradeoffs might be helpful to readers? Perhaps incorporated somehow into Fig 1 - which is a pretty figure, but does not add that much beyond the text. Fig 2 is more useful in terms of generating hypotheses.

This is a worthwhile point. In lieu of adding a figure, we have added some specific discussion about the trade-offs associated with microbial exposure from human-animal interactions when we introduce how microbial exposure could lead to changes in human health (Lines 201-205).

Moreover, there are trade-offs associated with microbial exposure during human-animal interactions as one can be exposed to not only to commensal or beneficial microbes but also potential pathogens. So, while exposure to microbes early in life can prime the immune system or provide metabolic functions, without other measures to combat infection (i.e., hygiene and sanitation), such exposure can also lead to childhood morbidities and mortalities.

In addition, both the “*Dichotomous Microbial Exposure*” and “*Changes in colonization resistance and disrupted immune training*” sub-sections of the “Changes in human health due to variation in microbial exposure” section explicitly presents findings related to benefits and costs of microbial exposure, illustrating potential trade-offs.

RIC5: Why do domestication and industrialization have similar shifts in microbial communities? Due to diet, housing, multispecies or other environmental interactions, stress? Since the authors are aiming to promote research in this important area, some hypotheses about factors influencing the similarities would be welcome. (mentioned on p10 and then factors are proposed on p.12 - perhaps connect them more closely in the text)

Variation in microbial exposure at the human-animal interface and the implications for microbiome-mediated health outcome

Thank you for pointing out the possibility of a more explicit treatment of hypotheses regarding similarities between domesticated animal and industrialized human population characteristics. We have now included some hypotheses for why domesticated animals and industrialized humans experience similar microbial shifts and frame these parallel shifts in the context of human-animal and animal-animal interactions (Lines 320-328).

Just as humans experience shifts in factors that impact the microbiome during industrialization, so do domesticated animals experience new diets, altered social structures, and different housing and environmental exposure relative to their wild progenitors. Importantly, there are parallels in how humans and domesticated animals interact with other animals across the gradient of industrialization. Wild animal contact is likely higher for both humans and domesticated animals in less industrialized contexts, whereas more industrialized and domesticated humans and animals likely interact with more companion animals. These differences will impact the frequency and diversity of microbial transmission via human-animal and animal-animal interactions.

RIC6: The discussion of suburban as opposed to rural-urban was interesting, and highlights environmental variation and exposure issues.

Thanks. We agree the suburban lifestyle provides an especially useful lens through which to consider environmental variation. Hopefully more research into populations of this type may be stimulated by our paper.

Reviewer 2 Suggestions (These are drawn from comments in the Reviewer provided track changes file. Additional line edits have also been integrated as appropriate.):

R2C1: Re: Figure 1. This beautiful figure should be better described in the figure legend so that a naïve reader can clearly understand the message. The authors seem to display differences in lifestyles by colors (light green section with huts for traditional HG/agrarian ecosystem; olive green for a rural industrialized context, grey seems to be indicating urban environment, and blue shows industrialized setting. Is that correct? Why the choice of animals, particularly monkeys in the industrialized setting? Clearly describing how this figure illustrates the “One Health” concept will be useful to readers.

We hope all readers will gain from the figure, so have updated the legend as the Reviewer suggests. It now reads:

Figure 1: Varied lifestyles promote differential interactions with animals, where not only the type of animal differs but so does the type of interaction. We hypothesize these distinct interactions shape microbial transmission across an environment and result in human gut microbiome variation. We illustrate some of the possible lifestyle contexts including Western industrial rural (top left quadrant); non-Western, non-industrialized or traditional rural (top right quadrant); non-Western industrializing city (bottom right quadrant); and Western industrialized city (bottom left quadrant). Each setting will have a unique composition of direct and indirect interactions among humans, animals, and their shared environment. The animals shown here typify those possible interactions but are not a comprehensive representation. For example, in non-Western industrializing cities, such as Mumbai, some wild animals, especially monkeys, have adapted to living there. While wild animals, like coyotes, can be found in Western industrialized cities, they are much less common, different species, and less likely to interact with humans directly, resulting in a unique human-animal interaction suite between the two environment types.

R2C2: Re: Lines 55-57. [Disentangling the microbiome effects of dichotomous microbial exposure at human-animal interfaces from other environmental and lifestyle factors] is hard to do in practice. Perhaps the authors could include a segment before the Conclusion section to describe a few ways this could be done?

We agree with the reviewer that this is hard, but it is an important goal. To help make it more feasible, we have added text to the “Future Directions” section that presents recommendations for future work (Lines 399-431).

...As such, sampling diverse, underrepresented populations in future studies is crucial to capture the range of microbiomes across the human species, and to specifically recognize the large variation of human-animal interface effects on the human microbiome (147, 149). One route to doing so is leverage ongoing One Health studies to characterize the gut microbiome of people for whom fecal samples, health information, and environmental covariate data is already available. Here, using already banked samples and implementing the practice of banking future samples will be crucial to gain access to data from a larger representation of human populations. Furthermore, utilizing field-friendly techniques such as storing fecal samples in ethanol may

Variation in microbial exposure at the human-animal interface and the implications for microbiome-mediated health outcome

open avenues for more research on human-animal interface effects in locales where freezer availability is limited. By considering rapidly industrializing populations, we can assess how recent changes in human-animal interactions along the gradient of industrialization may drive microbial spread and shifts in populations.

In addition to analyzing existing human samples, we should also focus on collecting samples from non-human hosts (i.e., animals, water, and soil) to better understand how pathogenic and commensal microbes cycle between humans, animals, and the environment. Creating social or interaction networks to map microbial transmission at human-animal interfaces could identify humans, animals, or environmental features that act as microbial super-spreaders. Collecting samples multiple times within a year will be important for identifying seasonal shifts in microbial exposure from host-associated and environmental microbial communities (150, 151).

Apart from sampling populations at a broader geographic and temporal scale, we could also benefit from using domesticated animals as observational comparisons and experimental models. Experimental manipulations with domesticated animals could help disentangle microbiome effects, and the resulting host impacts, of animal-associated microbial transmission from other environmental and lifestyle factors. These manipulations may also aid in determining the direction and dynamics of microbial transmission during host species interactions, potentially allowing the creation of predictive models of beneficial and pathogenic microbial transmission (53). Domesticated animals can be considered not only as examples for disparities in microbial exposure but also as leading players in future efforts to mitigate the health effects of altered microbial diversity in industrial human settings. Importantly, domesticated animals may serve as microbial “reservoirs” in otherwise depauperate environments and mitigate microbially-mediated diseases.

R2C3: Re: Lines 74-75. Not clear what the authors mean by “more as well as more types”. Do they mean higher microbial load and diverse microbe types/species?

To address the Reviewer’s and possible reader’s confusion we’ve amended this text (now Lines 78-79) to read:

Increased interactions between humans, animals, and the environment can lead to a greater volume and diversity of microbes moving between different host species.

R2C4: Re: Line 162-165. It is true that the association between drinking water and cooking fuel sources and the microbiome may be proxies for environmental/animal exposures. But we also replicated this finding in the Hadza hunter-gatherers. I am curious how the authors interpret this finding in the Hadza.

We recognize that our prior text did not sufficiently address the Hadza sampling in Jha et al. and Fragiadakis et al., well as the Nepalese results. We have updated the text (now Lines 169-171).

Other studies in Himalayan and Hadza populations (56, 70) found that drinking water and cooking method – both routes of direct microbial exposure and proxies for environmental and animal exposure - were associated with differences in gut microbial composition.

R2C5: Re: Lines 167-169. I am not sure how much of the unexplained portion of the gut microbial variation will be explained by integrating animal interaction. Some portion of unexplained gut microbial variation will certainly be lined to it but whether it will be large or small portion remains to be seen.

The Reviewer is certainly correct that not all unexplained variation will be attributable to human-animal interface variation. To address this point (as well as a similar one raised by Reviewer 1 in R1C3), we have edited the text to be clearer about what our expectations are (Lines 171-176).

However, it is worth noting that the highlighted variables in all these studies typically only explain approximately 30% of variation in microbial composition. It may be that impacts of microbial exposure at the human-animal interface are embedded partially in that 30%, or, if they were explicitly tested for, such impacts could play a role in explaining some of the remaining 70% of variation.

R2C6: Re: Line 206. Why are there no cats in Figure 1?

While cats would certainly be found in most if not all of the types of environments that we include in Figure 1, we only include a single cat in the Industrialized Rural quadrant. The confusion this might engender should be addressed by the legend update (see R2C1 above) which now explicitly states that the animals displayed in the figure are representative of the different environments and are not exhaustive.

R2C7: Re: before Conclusion. I recommend the authors to include a Figure and short description to describe how future studies can integrate One Health approach to integrate human-animal interactions. What should be sampled? Where? How often, etc. While the paper summarizes the current state of microbiome research well and presents several convincing arguments regarding the role animals play in shaping the human microbiome, it does not provide a clear recommendation on how future research studies can integrate these elements in their study design. Including a section to recommend some designs would be make this paper very helpful to readers.

To maximize the potential value of our manuscript to researchers and practitioners, we have updated the “Future directions” section to include recommendations for future work (Lines 399-431, see text copied in R2C2 above).

July 21, 2021

Dr. Aspen T. Reese
University of California, San Diego
La Jolla, CA

Re: mSystems00567-21R1 (Variation in microbial exposure at the human-animal interface and the implications for microbiome-mediated health outcome)

Dear Dr. Aspen T. Reese:

Your manuscript has been accepted, and I am forwarding it to the ASM Journals Department for publication. For your reference, ASM Journals' address is given below. Before it can be scheduled for publication, your manuscript will be checked by the mSystems senior production editor, Ellie Ghatineh, to make sure that all elements meet the technical requirements for publication. She will contact you if anything needs to be revised before copyediting and production can begin. Otherwise, you will be notified when your proofs are ready to be viewed.

As an open-access publication, mSystems receives no financial support from paid subscriptions and depends on authors' prompt payment of publication fees as soon as their articles are accepted. =

Publication Fees:

We recognize that the video files can become quite large, and so to avoid quality loss ASM

suggests sending the video file via <https://www.wetransfer.com/>. When you have a final version of the video and the still ready to share, please send it to Ellie Ghatineh at eghatineh@asmusa.org.

Sincerely,

Suzanne Ishaq
Editor, mSystems

Journals Department
Phone: 1-202-942-9338